# Evaluation of diagnostic measurements in patients with non-specific complaints: A secondary analysis after implementation of a care-pathway in the emergency department

M. G. A. M. van der Velde[1,2]*, M. A. C. Jansen[3], H. R. Haak[1,2], M. N. T. Kremers[2,4]

1 Department of Internal Medicine, Máxima MC, Veldhoven, the Netherlands, 2 Department of Health Services Research, and CAPHRI School for Public Health and Primary Care, Aging and Long Term Care Maastricht, Maastricht, the Netherlands, 3 Netwerk Acute Zorg Brabant, Tilburg, the Netherlands, 4 Emergency Department, Erasmus University Medical Center, Rotterdam, the Netherlands

* marleen.van.der.velde@mmc.nl

## Abstract

### Background

Older patients presenting with nonspecific complaints (NSC) in the Emergency Department (ED) pose diagnostic challenges. The lack of clear symptoms leads to high misdiagnosis rates, extended hospital stays, and functional impairment. However, limited research exists on diagnostic test utilization for this population.

### Methods

A secondary analysis was conducted on data from 399 older patients (aged 70+) at two hospitals in the Netherlands. These patients presented with NSC and were assessed using a standardized care-pathway that included diagnostic tests such as blood tests, ECGs, chest X-rays, and bladder ultrasounds. Data from a control group (164 patients) and an intervention group (235 patients) were compared, focusing on adherence to the diagnostic pathway, test frequency, and clinical outcomes.

### Results

The intervention group showed significantly greater use of several diagnostic tests compared to the control group, specifically calcium, TSH, glucose, ALAT, ECGs, chest X-rays, and bladder ultrasounds. Notably, abnormal findings were relatively low across tests, particularly for ALAT, TSH and calcium. Urinalysis, chest X-rays and ECGs were used more frequently and identified clinically significant findings. Head CT scans were used more frequently in the intervention group, though not statistically significant.

**Data availability statement:** The data cannot be shared publicly as they are licensed to the authors under a data use agreement with the participating hospitals. This restriction is in place to ensure compliance with ethical guidelines and agreements designed to protect patient privacy and confidentiality. The study was approved by the ethics committee at the Máxima Medical Centre in Veldhoven (ref. no N19.034) as well as by the participating hospitals. Researchers interested in accessing the pseudonymized data may submit a request to this METC by contacting the ethics committee at metc@mmc.nl.

**Funding:** The author(s) received no specific funding for this work.

**Competing interests:** The authors have declared that no competing interests exist.

## Conclusion

We recommend a standardized laboratory work-up for patients presenting with NSC. There is no justification for the routine use of ALAT, TSH and calcium measurements. We also recommend to incorporate bladder scans into routine care, and continuing the standardized use of ECGs, chest X-rays and urinalysis. Head CT scans, on the other hand, should be based on individualized clinical decisions.

**Trial number**: Dutch Trial register, number NL8960.

## Introduction

Patients presenting to the Emergency Department (ED) with nonspecific complaints (NSC) pose significant diagnostic challenges for physicians. NSC, such as fatigue, dizziness, or general malaise, often lack clear indications of underlying conditions, making it difficult for clinicians to quickly and accurately determine the cause [1–3]. Consequently, the rates of misdiagnosis are high, leading to serious consequences for patients such as prolonged hospitalization, functional impairment and mortality [3–6]. These patients might benefit from a systematic approach, as outlined in our integrated care-pathway, which includes risk stratification, systematic patient assessment and standard diagnostic measurements to ensure appropriate and timely interventions [7]. However, the use of standardized testing carries a risk of over-diagnosis, making the choice of appropriate diagnostic measures critical.

Limited knowledge exists about diagnostic resource utilization for patients with NSC. For example, Bhalla et al. reported a significant increase in diagnostic tests and procedures among older patients presenting with symptoms like weakness and fatigue [8]. However, these findings were not replicated in a similar cohort [3,9]. A recent systematic review highlighted the overall scarcity of data on resource utilization for NSC-patients in the ED [10].

Given this knowledge gap, implementing diagnostic tests requires a balance between accuracy, efficiency, and cost while minimizing false positives and unnecessary procedures [11]. Effective diagnostics can timely identify serious conditions, improving patient outcomes and streamlining ED flow. Identifying which tests provide the most benefit for NSC patients is key to develop more effective care-pathways, ultimately enhancing quality of care and resource management in the ED.

Therefore, we conducted a secondary analysis of data from two hospitals that implemented the NSC care-pathway. This analysis was part of a study in the Netherlands, which evaluated the effectiveness of the care-pathway in the ED for patients presenting with NSC. Our secondary analysis aimed to assess the feasibility of the diagnostic tests performed, identify any unnecessary tests, and determine whether additional testing within the NSC-pathway might be needed.

## Methods

### Design and setting

We conducted a before and after study in two hospitals in the Netherlands between April 2021 and November 2024; a teaching hospital (hospital 1) and a general hospital (hospital 2), both located in urban areas.

Hospital 1 provides 24/7 ED service and handles approximately 20,500 ED visits annually, experiencing presentation stops occasionally. The ED houses 19 operational bed during office-hours and 15 operational beds during evening and nights. Emergency Physicians (EP) and internal medicine residents are 24/7 present and both are involved in the care of ED patients. Internists are present during office hours and available on call during evenings and nights. Hospital 2, with a lower annual ED volume of about 13,500 visits, also offers 24/7 ED service and experiences infrequent presentation stops. The ED includes seven operational beds and four additional observation beds, intended for patient monitoring rather than admission, and supervised by the EP. EPs and internists are present during the day and evening shifts, while at night they are on call. EP residents are present around the clock, whereas internal medicine residents are only available during office hours.

### Population

The study included patients aged 70 and older who presented to the ED with NSC and met specific inclusion criteria. These criteria included referrals by general practitioners or elderly care physicians during weekday working hours, or triage in the ED with NSCs requiring ED admission. NSC were divided into five referral categories: 1) somatic problems, such as weakness, not feeling well, change in nutritional status or unexplained weight loss; 2) an increased demand of care, such as loss of independency, a necessity for starting home-care or admission to a care-home, not indicated previously; 3) cognitive problems, such as disorientation, changes in behavior, abrupt cognitive decline; 4) a decline in functional status, such as loss of mobility; and 5) unexplained falls, not related to extrinsic factors.

### NSC care-pathway

The detailed trial protocol, including specifics of the care-pathway, has been published previously [7,12]. Before the implementation of the care pathway, patients received standard care. Upon arrival, they underwent triage, followed by history taking and physical examination. Frailty screening was not routinely performed. Also, screening across the multiple domains of the comprehensive geriatric assessment (CGA) was left to the discretion of the attending physician. Diagnostic tests were ordered according to the physician's and nursing staff's judgment.

In contrast, the care pathway offered standardized care for patients with NSC [7]. Upon their arrival at the ED, patients were triaged, and assessed for frailty in both cognitive as functional domains using the APOP-screener [13]. In addition to solely somatic history taking, physicians conducted screening across various domains in line with the CGA [14]. NSC-patients received standard diagnostic measurements, including an electrocardiogram, urinalysis, chest radiography, bladder ultrasound and comprehensive blood tests (Table 1). Further diagnostic tests could be performed according to the physician's choice.

### Data-collection

Before implementing the care-pathway, one hospital used a graphic user interface data mining tool (CTcue, version 3.0) to create its control group [15]. This tool allowed unstructured electronic health record (EHR) data to be retrieved through specific queries. Data collection was conducted anonymously, adhering to predefined criteria and endpoints, eliminating the need for informed consent.

In the other hospital, where CTcue was unavailable, the control group was assembled manually by an emergency physician (EP), who registered eligible patients before the care-pathway's implementation and used pseudonymized data,

**Table 1. Prerequisite diagnostic measurements for NSC-patients.**

| Diagnostic measurements | Reference value – laboratory | Values deemed of clinical significance |
|---|---|---|
| **Laboratory diagnostics** | | |
| CRP | <5 mg/L | > 30 mg/L |
| Hemoglobin | Males 8.5–11.0 mmol/L | Males <7.0 mmol/L |
| | Females 7.5–10.0 mmol/L | Females <6.5 mmol/L |
| Thrombocytes | 150-400x109/L | <50 or >500x109/L |
| Leukocytes | 4.0-10.0 x109/L | <4.0 or >10.0 x109/L |
| Glucose | 4.0-7.8 mmol/l | <3.0 or >11.1 mmol/l |
| | | Hypoglycemia <3.0 |
| | | Hyperglycemia >11.1 |
| Sodium | 135-145 mmol/ | <130 or >150 mmol/l |
| Potassium | 3.5-5.0 mmol/l | Potassium <3.0 or >5.0 mmol/l |
| Estimated GFR | >90 ml/min/{1.73m2) | <30 ml/min/ {1.73m2} |
| ALAT | <45U/L | >100 U/L |
| GGT | <55U/L | >100 U/L |
| Calcium | 2.10-2.55 mmol/L | <2.00 or >2.90 mmol/L |
| TSH | 0.27-4.2 mmol/L | <0.01 or >20 mmol/L |
| **Urine-analysis** | Culture if analysis is suspect | Positive culture or; |
| | | Negative culture but did already lead to clinical actions. |
| **Electro cardiogram** | NA | Normal |
| | | Rhythm/conduction disorder (not pre-existent) |
| | | Signs of ischemia (not pre-existent) |
| | | Inconclusive |
| **Chest radiography** | NA | Normal |
| | | (Suspicion of) Infection/tumor related changes |
| | | Pleural effusion/edema |
| | | Fractures |
| | | Inconclusive |
| **Bladder ultrasound** | NA | Catheterization performed |

CRP, C-reactive protein; GFR, glomerular filtration rate – calculated by CKD-EPI; ALAT, alanine-aminotransferase; GGT, Gamma-glutamyl transpeptidase; TSH, thyroid stimulating hormone.

waving the need for informed consent. After implementing the care-pathway, eligible patients were provided written information at ED presentation and were asked to provide informed consent.

## Outcome measures

Data on prerequisite diagnostic measurements (Table 1), including additional tests such as COVID-19 polymerases chain reaction test (PCR), head computer tomography (CT) scans, were collected from the EHR.

Laboratory results were classified as either normal or abnormal, based on laboratory-specific reference values and expert opinions on clinically relevant abnormalities. Expert opinion was based on expertise of the authors (MV, HH, MK) and were deemed essential to provide a more context-sensitive interpretation of results, ensuring that relevant abnormalities were identified when necessary.

Urinalysis was classified as abnormal if a culture was performed, following an abnormal sediment finding. Abnormal urinalysis results were further categorized based on pathogen identification or treatment with antibiotics.

ECG results were compared to previous records and classified as normal, showing new rhythm/conduction disorders, or indicating new ischemia. For patients with pre-existing rhythm disorders, the ECG was categorized as normal if no new abnormalities were identified.

Chest X-rays were classified as normal or as showing suspected infection or tumor-related changes, pleural effusion/edema, visible fractures, or inconclusive findings.

Bladder ultrasound results were recorded as continuous variables. However, in many cases, only the bladder volume was documented, without specifying whether it was post-void or a random measurement. Consequently, cases were only classified as retention bladder if catheterization was performed.

The treating physician could request additional tests, which were also recorded and evaluated. COVID-19 test results were documented as either positive or negative. Head CT scans were classified as normal or abnormal, with the specific indication for each scan documented. Scans were considered abnormal if they showed new findings compared to prior imaging, revealed acute pathology, or included abnormalities with potential clinical relevance. These findings were not always acute in nature and could also include previously undetected chronic conditions, such as old ischemic lesions. Any additional radiological examinations beyond the prerequisite diagnostics were recorded but not evaluated.

Values deemed of clinical significance were assessed based on their value in confirming or ruling out disease. In evaluating the results, we relied on our expertise (MV, HH, MK).

## Statistical analysis

We calculated the diagnostic tests performed for all NSC patients (both intervention and control groups) to assess adherence to the care-pathway and to evaluate its added value compared to standard care. Additionally, we analyzed the frequency of abnormal results, considering both deviations from reference values and their clinical significance.

Statistical analyses were conducted using SPSS, version 27 (IBM Corp, New York, USA). Nominal data were presented as total counts and percentages, with p-values calculated using Pearson's chi-square or Fisher's exact test for dichotomous variables. A significance level of 5% was applied to all statistical tests.

## Ethics

This study was approved by the ethics committee at the Máxima Medical Centre in Veldhoven (ref. no N19.034) and by participating hospitals. The ethics committee judged the pathway as standard care, therefore patients eligible for the pathway were included and were asked for informed consent at the end of the ED-visit. All participants gave their written informed consent prior to data collection. Dutch Trial register, number NL8960.

## Results

Hospital 1 included 128 control patients between August 1, 2020 and March 31, 2021 and 212 intervention patients between April 1, 2021 and November 30, 2024. Hospital 2 included 36 control patients between December 1, 2020 and August 31, 2021 and 23 intervention patients from September 1, 2021 until November 30, 2024. In total, 399 patients were included in the participating hospitals, comprising 164 control patients and 235 intervention patients.

## Baseline characteristics

Baseline characteristics are presented in Table 2. Age and sex were comparable between both control and intervention groups. Significant differences were found between the control and intervention group. Patients in the intervention group were more likely to use multiple medications, with 71.1% using five or more compared to 50.6% in the control group

**Table 2. Baseline characteristics.**

| | Control (n = 164) | Intervention (n = 235) | p-value |
|---|---|---|---|
| Age (mean ± SD) | 82.9 ± 5.9 | 83.9 ± 6.3 | 0.135 |
| **Sex (n (%))** | | | 0.761 |
| Male | 80 (48.8%) | 111 (47.2%) | |
| Female | 84 (51.2%) | 125 (52.8%) | |
| **Number of medication (n (%))** | | | <0.001 |
| 0 | 29 (17.7%) | 7 (3.0%) | |
| 1-2 | 26 (15.9%) | 20 (8.5%) | |
| 3-4 | 26 (15.9%) | 41 (17.4%) | |
| ≥ 5 | 83 (50.6%) | 167 (71.1%) | |
| **Mode of transportation (n (%))** | | | 0.824 |
| Ambulance | 80 (49.1%) | 117 (50.2%) | |
| Own transport | 83 (50.9%) | 116 (49.8%) | |
| **Specialist in charge (n (%))** | | | <0.001 |
| Emergency Physician | 53 (32.3%) | 36 (15.3%) | |
| Internal Medicine | 83 (50.6%) | 176 (74.9%) | |
| Geriatrician | 23 (14.0%) | 23 (9.8%) | |
| Other | 5 (3.0%) | 3 (1.3%) | |
| **Triage code (n (%))** | | | 0.019 |
| U0 | 1 (0.6%) | 0 (0.0%) | |
| U1 | 11 (6.8%) | 6 (2.6%) | |
| U2 | 58 (35.8%) | 63 (27.0%) | |
| U3 | 82 (50.6%) | 134 (57.5%) | |
| U4 | 4 (2.5%) | 14 (6.0%) | |
| U5 | 6 (3.7%) | 16 (6.9%) | |
| **Referral category (n (%))** | NA | | NA |
| Somatic | | 112 (47.7%) | |
| Higher need for care/social situation | | 22 (9.4%) | |
| Functional status | | 21 (8.9%) | |
| Cognitive problems | | 52 (22.1%) | |
| Inexplicable falling | | 28 (11.9%) | |

*Data are reported as number of participants (%), mean ± standard deviation or median (Q1-Q3). Percentages were computed relative to the total number of participants in the presented column. SD: standard deviation. GP: general practitioner. ECP: elderly care physician. NA: not applicable.*

(p < 0.001). They were also more frequently assigned lower triage levels (U3–U5: 70.4% vs. 56.8%, p < 0.001). The most common reason for referral in the intervention group was somatic problems (47.7%), followed by cognitive issues (22.1%).

### Diagnostic measurements – Adherence and feasibility

As shown in Table 3, a substantial proportion of the required laboratory diagnostic tests was already performed in the control group with completion rates ranging from 96.5% to 100%. Although glucose and alanine aminotransferase (ALAT) already had high completion rates in the control group, they were performed significantly more often in the intervention group (ALAT 98.6% vs. 93.9%, p < 0.001; glucose 99.6% vs 93.9%, p = 0.001). The largest differences were observed for calcium and thyroid stimulating hormone (TSH), which were measured significantly more frequently in the intervention group, with completion rates of 95.8% vs. 64.0% (p < 0.001) and 92.3% vs. 40.9% (p < 0.001), respectively.

**Table 3. Diagnostic measurements, performed.**

|  | Total (n = 399) | Control (n = 164) | Intervention (n = 235) | p-value |
|---|---|---|---|---|
| **Laboratory diagnostics** |  |  |  |  |
| CRP | 397 (99.5%) | 162 (98.8%) | 235 (100.0%) | 0.168 |
| Hemoglobin | 398 (99.7%) | 163 (99.4%) | 235 (100.0%) | 0.411 |
| Leukocytes | 398 (90.7%) | 163 (99.4%) | 235 (100.0%) | 0.411 |
| Thrombocytes | 396 (99.2%) | 163 (99.4%) | 233 (99.1%) | 1.000 |
| Glucose | 388 (97.2%) | 154 (93.9%) | 234 (99.6%) | 0.001 |
| Sodium | 398 (99.7%) | 163 (99.4%) | 235 (100.0%) | 0.411 |
| Potassium | 398 (99.7%) | 163 (99.4%) | 235 (100.0%) | 0.411 |
| Estimated GFR | 397 (99.5%) | 162 (98.8%) | 235 (100.0%) | 0.168 |
| ALAT | 370 (96.6%) | 153 (93.9%) | 217 (98.6%) | 0.011 |
| GGT | 372 (95.6%) | 154 (94.5%) | 218 (96.5%) | 0.346 |
| Calcium | 329 (82.5%) | 105 (64.0%) | 224 (95.3%) | <0.001 |
| TSH | 284 (71.2%) | 67 (40.9%) | 217 (92.3%) | <0.001 |
| **Urine-analysis** | 332 (84.1%) | 119 (72.6%) | 214 (92.2%) | <0.001 |
| **ECG** | 324 (81.2%) | 121 (73.8%) | 203 (86.4%) | 0.002 |
| **Chest radiography** | 339 (85.0%) | 119 (72.6%) | 22 (93.6%) | <0.001 |
| **Bladder ultrasound** | 215 (54.3%) | 51 (31.3%) | 164 (70.4%) | <0.001 |
| **Additional tests** |  |  |  |  |
| COVID test | 155 (38.8%) | 69 (42.1%) | 86 (36.6%) | 0.112 |
| CT-head* | 94 (23.7%) | 31 (19.1%) | 63 (26.8%) | 0.077 |
| Other† | 67 (16.8%) | 17 (10.4%) | 50 (21.3%) | 0.023 |

Values are presented in n (%). CRP: C-reactive protein. GFR: glomerular filtration rate – calculated by CKD-EPI. ALAT: alanine-aminotransferase. GGT: Gamma-glutamyl transpeptidase. TSH: thyroid stimulating hormone. ECG: electrocardiogram. CT: computer tomography.

* Trauma as reason for CT-head: total n = 53 (56.4%%), control n = 21 (67.7%), intervention n = 32 (50.8%).

† Other diagnostic measurements include: ultrasound, X-rays other than thorax, and CT-scans other than CT cerebrum.

All prerequisite diagnostic measurements, including urinalysis, ECG, chest X-ray, and bladder ultrasound, were performed significantly more frequently in the intervention group than in the control group (urinalysis: 92.2% vs. 72.6%, *p*<0.001; ECG: 86.4% vs. 73.8%, *p*=0.002; chest X-ray: 93.6% vs. 72.6%, *p*<0.001; bladder ultrasound: 70.4% vs. 31.3%, *p*<0.001).

COVID-19 tests were conducted only when clinically indicated, with no significant difference between the groups (36.6% vs. 42.1%, p=0.112). The use of head computed tomography (CT) scans was higher in the intervention group, though this difference was not statistically significant (26.8% vs. 19.1%, p=0.077). Notably, the proportion of head CTs performed due to trauma was lower in the intervention group than in the control group (50.8% vs. 67.7%).

Additional diagnostic examinations, such as other X-rays, ultrasounds, or CT scans (excluding head CT), were infrequently performed in both groups. However, a significantly higher percentage of these tests was conducted in the intervention group (21.3% vs. 10.4%, p=0.023).

In terms of laboratory diagnostics, adherence to the care-pathway was high, with TSH testing showing the lowest adherence at 92.3%. However, adherence to the care-pathway for additional diagnostics was less consistent, with bladder ultrasound being the most frequently omitted test, completed in only 70.4% of cases.

## Diagnostic measurements – Evaluation

The laboratory results from 399 patients, as shown in Table 4, revealed several trends. For example, C-reactive protein (CRP) showed the highest rate of abnormal findings (80.5%), but only 48.6% of these were deemed as clinically

**Table 4. Laboratory diagnostics, evaluation.**

| | Total (n = 399) | Outside of reference value | Outside of values deemed of clinical significance |
|---|---|---|---|
| **CRP** | 397 (99.5%) | 318 (80.5%) | 193 (48.6%) |
| **Hemoglobin** | 398 (99.7%) | | |
| Male | | 132 (69.1%) | 98 (24.6%) |
| Female | | 83 (39.9%) | 58 (14.5%) |
| **Leukocytes** | 398 (90.7%) | 164 (41.2%) | 164 (41.2%) |
| **Thrombocytes** | 396 (99.2%) | 86 (21.7%) | 13 (3.3%) |
| **Glucose** | 388 (97.2%) | 123 (30.8%) | 47 (12.1%) * |
| **Sodium** | 398 (99.7%) | 128 (32.3%) | 36 (9.0%) |
| **Potassium** | 398 (99.7%) | 64 (16.1%) | 36 (9.0%) |
| **Estimated GFR** | 397 (99.5%) | 361 (90.9%) | 57 (14.4%) |
| **ALAT** | 370 (96.6%) | 48 (13.0%) | 11 (3.0%) |
| **GGT** | 372 (95.6%) | 88 (23.7%) | 37 (9.9%) |
| **Calcium** | 329 (82.5%) | 54 (16.4%) | 15 (4.6%) |
| **TSH** | 284 (71.2%) | 49 (17.3%) | 7 (2.5%) |

Values are presented in n (%). CRP, C-reactive protein; GFR, glomerular filtration rate – calculated by CKD-EPI; ALAT, alanine-aminotransferase; GGT, gamma-glutamyl transpeptidase; TSH, thyroid stimulating hormone.

Reference values are: CRP < 5 mg/L. Hemoglobin males 8.5–11.0 mmol/L, females 7.5–10.0 mmol/L. Leucocytes 4.0–10.0 x10$^9$/L. Thrombocytes 150-400x10$^9$/L. Glucose 4.0–7.8 mmol/l. Sodium 135–145 mmol/l. Potassium 3.5–5.0 mmol/l. eGFR: CKD-EPI > 90 ml/min/ {1.73m$^2$}. ALAT: < 45U/L. GGT < 55U/L. Calcium 2.10–2.55 mmol/L. TSH 0.27–4.2 mmol/L.

Values of clinical significance are: CRP > 30 mg/L. Hemoglobin males <7.0 mmol/L, females <6.5 mmol/L. Leucocytes <4.0 or >10.0 x10$^9$/L. Thrombocytes <50 or>500x10$^9$/L. Glucose <3.0 or >11.1 mmol/l. Hypoglycemia <3.0, Hyperglycemia >11.1. Sodium <130 or >150 mmol/l. Potassium <3.0 or >5.0 mmol/l. eGFR: CKD-EPI < 30 ml/min/{1.73m$^2$}. ALAT: > 100 U/L. GGT > 100 U/L. Calcium <2.00 or >2.90 mmol/L. TSH < 0.01 or >20 mmol/L.

*No hypoglycemia's were observed.

significant. Similarly, notable clinically significant abnormalities were observed in hemoglobin (14.5–24.6%), glucose (12.1%), and sodium (9.0%). Estimated glomerular filtration rate (eGFR) had a high abnormality rate (90.9%), yet only 14.4% were considered clinically significant. ALAT, GGT, and calcium had relatively low rates of clinically significant abnormalities (3.0%, 9.9%, and 4.6%, respectively). Thyroid screening had the lowest testing frequency (71.2%) and the fewest clinically significant findings (2.5%).

Urinalysis was conducted in 331 patients (83.8%) (Table 5). Among these, 212 patients (64.0%) had normal urinalysis results, and no urine culture was performed. Of the urine cultures that were conducted, 66 patients (19.9%) had an identified pathogen, 29 patients (8.8%) showed mixed flora and were treated with antibiotics, while 24 patients (7.3%) had mixed flora but did not receive antibiotic treatment. ECGs revealed new rhythm/conduction disorders in 13.3% of patients and ischemia in 2.2%. Chest X-rays identified suspected pneumonia or malignancy in 16.4% of cases and pleural effusion or edema in 5.3%. Bladder ultrasounds showed urine retention in 28.0% of patients, while head CT scans, revealed abnormalities in 21.3% of cases.

## Discussion

We assessed the impact of a standardized diagnostic approach for older patients with NSC presenting in the ED. Our goal was to evaluate the feasibility of standardized diagnostic tests for this complex patient population and identify unnecessary ones.

We found that the introduction of the diagnostic care-pathway resulted in significant changes in clinical practice with adequate adherence to the protocol. Some blood tests, including calcium, TSH, glucose and ALAT, were performed more frequently in the intervention group. Additionally, the use of prerequisite diagnostic measures such as urinalysis, ECGs,

**Table 5. Other diagnostic measurements, evaluation.**

| | Total (n = 399) | Abnormal |
|---|---|---|
| **Urine-analysis** | 331 (83.8%%) | |
| *Pathogen identified* | | 66 (19.9%) |
| *Mixed flora, treated with antibiotics* | | 29 (8.8%) |
| *Mixed flora, not treated with antibiotics* | | 24 (7.3%) |
| **ECG** | 324 (81.2%) | 52 (16.0%) |
| *Rhythm/conduction disorders* | | 43 (13.3%) |
| *Ischemia* | | 7 (2.2%) |
| *Inconclusive* | | 2 (0.6%) |
| **Chest radiography** | 340 (85.2%) | 79 (23.2%) |
| *Suspected pneumonia/tumor* | | 56 (16.4%) |
| *Pleural effusion/edema* | | 18 (5.3%) |
| *Fractures* | | 4 (1.2%) |
| *Inconclusive* | | 1 (0.3%) |
| **Bladder ultrasound\*** | 215 (54.3%) | 68 (28.0%) |
| **COVID test** | 155 (38.8%) | 27 (17.4%) |
| **CT-cerebrum** | 94 (23.7%) | 20 (21.3%) |
| **Other (n (%))** | 67 (16.8%) | |
| X-ray (other than thorax) | 31 (7.8%) | NA |
| Ultrasound | 18 (4.5%) | NA |
| CT (other than cerebrum) | 18 (4.6%) | NA |

ECG: electrocardiogram. CT: computer tomography. \* In 19 patients (4.8%) no ultrasound was performed as a catheter was already in use.

chest X-rays, and bladder ultrasounds increased significantly in the intervention group. Although additional imaging tests (e.g., X-rays, ultrasounds, CT scans) were used infrequently overall, their use was more common in the intervention group. This pattern suggests that these tests may have been ordered when initial diagnostics proved inconclusive. However, this observation raises important considerations. On one hand, the lower use of imaging in the control group may point to a risk of underdiagnosis, where potentially critical conditions could be missed due to insufficient diagnostic efforts. On the other hand, the increased use of imaging in the intervention group could contribute to overdiagnosis, where incidental findings, unrelated to the patient's complaints, are identified and may result in unnecessary follow-ups or interventions [16].

Adherence to the laboratory components of the diagnostic pathway was generally high. This was likely due to the use of pre-set lab panels in the Electronic Health Record (EHR), which made it easier to order tests efficiently upon patient arrival, However, it is important to note that most EDs use pre-set lab panels for various presenting complaints, regardless of this specific care pathway. In contrast, tests requiring more effort, such as ECGs and bladder ultrasounds, were more likely to be omitted. This may reflect practical barriers, including time constraints and resource limitations. Further exploration is needed to determine the clinical impact of these omissions on patient outcomes.

Despite the increased use of laboratory testing, the clinical utility of these tests remains uncertain. The prerequisite laboratory tests align with expert recommendations in the comprehensive geriatric assessment guidelines, offering a complete picture of inflammation, blood count, fluid and electrolyte balance, and liver, kidney, and thyroid function [17]. In our study, the proportion of clinically significant abnormal results was low, prompting the question whether these tests should be routinely included in diagnostic protocols within NSC patients or only used when indicated based on the patient's

medical history or physical examination. Research indicates that routine electrolyte testing, including calcium, magnesium, and phosphorus, rarely identifies clinically significant abnormalities or alters patient management [18,19]. This aligns with our findings, where abnormal results calcium-levels were infrequent. Similarly, previous research suggests that TSH abnormalities in older adults are typically transient and have minimal impact on functioning and should therefore be used for targeted assessments of suspected thyroid disorders rather than routine screening in undifferentiated cases [20–22]. ALAT was abnormal in only 3.0% of cases in our study, consistent with findings from a U.S. study demonstrating a similarly low diagnostic yield of routinely ordered liver tests [23]. ALAT is frequently included as part of automated biochemical panels, yet our results suggest its clinical utility in patients presenting with nonspecific complaints (NSC) is limited. Based on the low proportion of abnormal results, it may be reasonable to consider excluding ALAT from standardized diagnostic protocols for this population. However, this is complicated by the fact that gamma-glutamyl transferase (GGT) was elevated in 9.0% of cases in our cohort. As isolated GGT elevations often require contextual interpretation alongside transaminase levels, we recommend using GGT as a screening parameter, with ALAT re-evaluated only when GGT is abnormal. In light of its limited added value, we suggest removing ALAT from the standardized panel at this stage. This approach warrants further evaluation and may be adjusted as broader evidence emerges.

Based on these findings, we recommend adhering to our prespecified laboratory work-up for NSC patients, while excluding standardized measurements of ALAT, calcium, and thyroid screening—unless medical history, physical examination, or other laboratory values indicate their relevance.

Urinalysis, performed more frequently in the intervention group, detected bacteriuria in about 20% of cases. In nearly 10%, antibiotics were prescribed for what appeared to be asymptomatic bacteriuria. While clinical judgment likely informed these decisions, we did not monitor whether signs of infection were present. Diagnosing a urinary tract infection (UTI) in older adults is complex, particularly in patients with NSC, as presentations are often atypical, such as confusion, falls, or reduced appetite, rather than classic urinary symptoms [24]. Genitourinary infections are among the most frequent diagnoses in this group, which may justify performing urinalysis as part of the diagnostic work-up [3,25]. However, given the high prevalence of asymptomatic bacteriuria and the risk of false positives, urinalysis should always be interpreted in the broader clinical context [26]. While our findings support routine urinalysis in this group, treatment decisions should not rely solely on these results. Instead, urinalysis should be one component of the overall diagnostic reasoning, helping to assess the likelihood of a UTI. It is also important to consider waiting for culture results and to explore alternative explanations for symptoms to avoid unnecessary antibiotic use.

The diagnostic value of chest X-rays in the ED remains a topic of debate. While several studies have questioned the usefulness of routine chest X-rays, particularly in the absence of respiratory symptoms, these studies often excluded patients with unreliable or incomplete medical histories, such as those with cognitive impairment [27,28]. As a result, their findings are less applicable to older adults presenting with NSC, in whom history taking is frequently limited and atypical, minimally symptomatic presentations are more common. In such cases, chest X-rays can help detect otherwise unrecognized conditions like pneumonia or heart failure, which often present atypically. Given the diagnostic uncertainty and broad differential diagnoses characterizing NSC presentations, a standardized approach that includes chest X-rays may contribute to more accurate and timely diagnoses. Point-of-care ultrasound (POCUS) is increasingly recognized as a rapid, low-risk diagnostic tool with significant potential in geriatric emergency care [29,30]. At the time our care pathway was implemented, POCUS was not yet routinely used in the participating EDs, but its growing adoption suggests it could become a valuable alternative to chest X-rays in evaluating older adults with NSC.

It could also be argued that urinalysis or chest X-rays should be reserved for cases with specific indications, such as elevated inflammatory markers, signs of heart failure, or urinary retention. This targeted approach aligns diagnostics with clinical suspicion, minimizing unnecessary testing and enhancing relevance. However, it may also delay decision-making and prolong ED length of stay, complicating patient flow. Notably, Wachelder et al. identified neoplasms, urogenital, endocrine/metabolic, and respiratory diseases among the top five diagnoses in NSC patients presenting to the ED [3]. Given

this knowledge, we propose adherence to a more standardized approach, including routine urinalysis and chest X-rays, to rule out these common and potentially serious conditions, ensuring that critical diagnoses are not overlooked in the complex NSC-population.

Bladder ultrasound demonstrated clear clinical value given the presence of bladder retention in 28% of the cases and therefore should be part of the standard diagnostic protocol for NSC patients.

Our findings showed that 16% of NSC-patients showed significant abnormalities on ECG. This indicates that the diagnostic yield of ECG in this population is relatively high, with the test revealing a potentially important abnormality in approximately 1 out of every 6 patients. Given the often vague and non-specific nature of NSC presentations and the frequent unreliability of history taking in this population, the ECG appears to be a valuable diagnostic tool that can uncover underlying cardiac conditions that might otherwise remain undetected. These results support the routine use of ECG in the initial assessment of patients with NSC, as it may contribute to earlier recognition of serious pathologies and guide further diagnostic or therapeutic steps.

CT imaging of the head was performed more frequently in the intervention group, suggesting that head CT was used when the diagnostic pathway did not clarify the clinical presentation. However, this increase should be interpreted with caution due to potential selection bias, as CT was not a standard protocol. Preexisting indications, such as suspected bleeding, may have influenced its more frequent use. Due to this selection bias, we cannot recommend routine inclusion of head CT in the care -pathway. We advise following mild traumatic brain injury guidelines and considering CT for unexplained new-onset cognitive or behavioral changes.

Balancing a targeted diagnostic approach with efficiency of care is essential for optimizing resource use and improving patient outcomes in the ED. However, when evaluating the value of diagnostic testing, it is important to also consider the significance of normal results in this specific population. In NSC patients, clinical history and examination often provide limited clues and a broad differential diagnosis is common, normal test results can be particularly valuable. These results help to rule out potential diagnoses, allowing clinicians to narrow their diagnostic focus and refine their decision-making process, ultimately improving diagnostic accuracy and efficiency in a complex clinical environment.

In summary, we recommend following our standardized laboratory work-up, excluding ALAT, TSH and calcium, incorporating bladder scans into routine care, and continue the standardized use of chest X-rays, urinalysis and ECGs. CT scans of the head should remain clinical decisions made on an individual basis. For our recommended work-up, see Table 6.

**Table 6. Recommended diagnostic measurements for NSC-patients.**

| Diagnostic measurements | |
|---|---|
| Laboratory diagnostics | **Urine-analysis** |
| CRP | **Chest radiography** |
| Hemoglobin | **Bladder ultrasound** |
| Thrombocytes | **ECG** |
| Leukocytes | |
| Glucose | |
| Sodium | |
| Potassium | |
| Estimated GFR | |
| GGT | |

CRP, C-reactive protein; GFR, glomerular filtration rate – calculated by CKD-EPI; GGT, gamma-glutamyl transpeptidase; ECG; electrocardiogram.

## Strengths and limitations

This is the first study to assess specific diagnostic tests in older adults with nonspecific complaints (NSC), offering valuable insights into the use of diagnostic measures for this complex population. By standardizing diagnostic measurements, the results of this study could guide future clinical practice and guidelines for managing NSC patients in the emergency department. Limitations of this study are that as a secondary analysis, the findings should be considered preliminary and hypothesis-generating. Additionally, the retrospective nature of the control group could have introduced selection bias, but this is deemed to have minimal impact, as two-thirds of the participants were included prospectively, strengthening the overall reliability of the findings.

We recommend that future research involve a prospective, multicenter cohort study with a larger sample size, where NSC patients are included upon ED admission. This study should follow the recommendations of our diagnostic care pathway to evaluate its reliability in different NSC categories—such as somatic or cognitive problems or recurrent falls—and to further refine and personalize the diagnostic pathways. Additionally, it would be valuable to monitor how the results influence clinical decision-making and assess the impact of incidental findings. This approach could provide further insights into optimizing diagnostic strategies for NSC patients in the ED.

## Conclusion

In conclusion, the introduction of a standardized diagnostic pathway for older patients with NSC in the ED led to significant changes in clinical practice. We recommend a standardized laboratory work-up for patients presenting with NSC. There is no justification for the routine use of ALAT, TSH and calcium measurements. We also recommend to incorporate bladder scans into routine care, and continuing the standardized use of chest X-rays, urinalysis and ECGs. Head CT scans, on the other hand, should be based on individualized clinical decisions. Future research should focus on evaluating the impact of additional testing on patient outcomes and refining diagnostic approaches for specific NSC subcategories, such as somatic complaints, cognitive issues, or recurrent falls. These efforts will help optimize the care pathway for this complex patient population.

## Author contributions

**Conceptualization:** H.R. Haak.

**Data curation:** M.G.A.M. van der Velde.

**Formal analysis:** M.G.A.M. van der Velde.

**Methodology:** M.G.A.M. van der Velde, M.A.C. Jansen, H.R. Haak, M.N.T. Kremers.

**Project administration:** M.G.A.M. van der Velde.

**Supervision:** M.A.C. Jansen, H.R. Haak, M.N.T. Kremers.

**Writing – original draft:** M.G.A.M. van der Velde.

**Writing – review & editing:** M.A.C. Jansen, H.R. Haak, M.N.T. Kremers.

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
