## [Decision Letter · Decision Letter 0]

3 Jun 2025

Dear Dr. van der Velde,

 In particular, would you address the important observations made by Reviewer 2, especially your conclusions about urine analysis and ECG. You may feel that the suggestions for further analysis of additional outcomes made by Reviewer 1 would best be considered in a separate paper. Can you also check the phrasing on lines 166-169 that I found a bit confusing. The references also need reviewing to ensure they are complete - I spotted missing page numbers (ref 1), incorrect first author (ref 19) and lack of consistency of citing full or abbreviated journal names (e.g. refs 1 & 2). Has reference 12 now ben published? 

We look forward to receiving your revised manuscript.

Kind regards,

Antony Bayer

Academic Editor

PLOS ONE

2. In the online submission form, you indicated that [Data is available upon reasonable request. Researchers can contact the corresponding author for further details or to request access to the datasets used in this study.].

4. We note you have included a table to which you do not refer in the text of your manuscript. Please ensure that you refer to Table 2 in your text; if accepted, production will need this reference to link the reader to the Table.

Additional Editor Comments (if provided):

Reviewers' comments:

Reviewer's Responses to Questions

**Comments to the Author**

1. Is the manuscript technically sound, and do the data support the conclusions?

Reviewer #1: Yes

Reviewer #2: Yes

2. Has the statistical analysis been performed appropriately and rigorously?

Reviewer #1: No

Reviewer #2: Yes

3. Have the authors made all data underlying the findings in their manuscript fully available?

Reviewer #1: Yes

Reviewer #2: Yes

4. Is the manuscript presented in an intelligible fashion and written in standard English?

Reviewer #1: Yes

Reviewer #2: Yes

Reviewer #1: This study addresses an important and timely issue in emergency medicine, optimizing the diagnostic approach for patients presenting with non-specific complaints. Given the growing number of older and medically complex patients in the ED, the evaluation of structured diagnostic pathways is highly relevant. The authors tackle a clinically significant challenge and contribute to an area where evidence is still evolving, making this research both valuable and of broad interest to clinicians and researchers alike.

The aim of the study, to assess feasibility, identify unnecessary tests, and evaluate the added value of the NSC care pathway, is clearly stated and relevant. However, the statistical approach, while appropriate for descriptive comparisons (i.e., counts, percentages, chi-square/Fisher’s exact tests), appears limited in its ability to fully address these broader evaluative goals. To strengthen the analysis, the authors might consider including:

- Multivariable analysis to adjust for potential confounders (age, sex, comorbidity, triage level) when comparing testing patterns or outcomes.

- Logistic regression to assess whether specific diagnostic tests were independently associated with relevant clinical outcomes (diagnosis accuracy, admission, return visits).

- If resource utilization is a focus, cost-effectiveness or time-to-diagnosis analyses could also enhance the study’s impact. Did it speed up decisions? Shorten length of stay in the ED? Reduced unnecessary admissions?

Adding these elements could provide a more robust evaluation of the pathway’s effectiveness and clinical relevance beyond adherence and test frequency.

Reviewer #2: This is an interesting study on a very important topic and directly relevant to clinical practice.

I have the following minor recommendations:

I think it would be helpful to list the eligibility criteria for the NSC - I had to look this up in the research protocol which was time consuming yet the reader does need to know definitions of NSC to be able to interpret this paper.

Please can you clarify what you mean by ‘normal or abnormal’ with respect to CT head - CT head is frequently abnormal in older patients but this is rarely acutely relevant in an ED situation - do you mean that there are new, acute changes or other chronic abnormal changes? The presence of acute changes directly relevant to the ED presentation is a significant finding so should be clarified as such - I would consider 21.3% of CT brains to have acute abnormalities to be very high in this setting, which could potentially alter your conclusions.

I am concerned that the paper recommends routine urinalysis in this population. Presence of a urinary pathogen does not necessarily constitute urinary tract infection (defined as ‘true infections’ in this paper) and may represent asymptomatic bacteria, of which there is a high prevalence in this population. I think your discussion suggests 30% of cases who had urinalysis received antibiotics? UTI should only be diagnosed in the presence of other clinical factors, not simply on urinalysis alone; even in non specific presentations there needs to be other signs or symptoms of UTI for diagnosis - are you able to elaborate if those that received antibiotics had them because of other signs or purely on urinalysis results? Please can the discussion reflect this: although there is some discussion around the risk of asymptomatic bacturia currently the paper recommends routine urinalysis - I do not feel recommending routine urinalysis for all patients is the natural conclusion of this paper without further study, and could potentially be harmful.

You mention that calcium and TSH should be excluded due to low yield of abnormal results with clinical significance, however there is a similar low yield for ALAT (3%). Can you explain why you have not suggested this be excluded? In the UK LFTs are not routinely measured in urgent care unless clinically felt necessary (and GGT only rarely according to clinical need) so this data is very helpful in this setting and could help reduce healthcare costs.

You mention that ECGs only rarely showed clinically significant abnormalities and should be used selectively, however in the data 16% of patients had a clinically significant abnormality, which I would consider frequent, rather than rare. I would suggest your data actually suggests ECG is very useful in this population - picking up abnormalities ion 1 in 6 patients, and should be routine in such cases.

The paper is otherwise well written and covers a very relevant and important area of healthcare. I suggest the above modifications to the conclusions/discussion to allow a more considered and clinically relevant interpretation of your results. Many thanks

**Do you want your identity to be public for this peer review?** For information about this choice, including consent withdrawal, please see our Privacy Policy

Reviewer #1: No

Reviewer #2: No

---

## [Author Response · Author response to Decision Letter 1]

4 Jul 2025

Point to point response

Editor:

Dear Dr. van der Velde,

Thank you for submitting your manuscript to PLOS ONE. After careful consideration, we feel that it has merit but does not fully meet PLOS ONE’s publication criteria as it currently stands. Therefore, we invite you to submit a revised version of the manuscript that addresses the points raised during the review process.

Dear Dr. Bayer,

We thank you for your effort to assess our manuscript. We greatly appreciate the constructive insights that have contributed to its improvement.

In response to your and the reviewers’ comments, we have revised the manuscript accordingly, with all changes highlighted using track changes. We have also provided a detailed, point-by-point reply addressing each of your concerns.

In particular, we have expanded the methods section to enhance this article as a stand-alone piece. We added details about the participating hospitals, emergency departments, and the criteria for nonspecific complaints (NSC) to ensure the study is fully self-contained and its results can be interpreted in light of these characteristics. Additionally, baseline characteristics were included to provide better insight into the study population.

We took note of the insightful comments regarding our recommended diagnostic work-up. Specifically, concerning the role of ECG, we acknowledge that a 16% rate of clinically significant abnormalities is notable and supports the routine use of ECG in this population. Regarding urinalysis, we appreciate the complexity of diagnosing urinary tract infections (UTIs) in older adults, especially those with NSC. While urinalysis contributed to detecting true infections, we emphasize that a positive urinalysis alone should not lead to antibiotic treatment without corroborating clinical signs or symptoms. We have clarified that urinalysis should be integrated with comprehensive clinical assessment and culture results to avoid overtreatment and unnecessary antibiotic use. This nuanced approach better reflects current evidence and clinical practice challenges in this population.

We believe that the revised manuscript is now suitable for publication in European Geriatric Medicine.

On behalf of all authors,

Marleen van der Velde

Comment 1.1: In particular, would you address the important observations made by Reviewer 2, especially your conclusions about urine analysis and ECG.

Reply 1.1: We thank Reviewer 2 for their insightful observations regarding our conclusions on urine analysis and ECG use in older adults presenting with nonspecific complaints (NSC). We have carefully reconsidered these aspects in light of their comments and revised our manuscript accordingly.

Regarding ECG, we acknowledge that our original wording may have understated its diagnostic value. With 16% of patients showing clinically significant abnormalities, this is indeed a notable yield—approximately one in six patients. Given the non-invasive nature, low cost, and potential to uncover actionable cardiac pathology, we now support the routine use of ECG in the initial assessment of NSC patients. This approach can facilitate earlier recognition of serious conditions that might otherwise remain undetected, especially considering the often vague presentations and challenges in history taking in this population.

Concerning urine analysis, we appreciate the complexity of diagnosing urinary tract infections (UTIs) in older adults, especially those with NSC, due to the high prevalence of asymptomatic bacteriuria and atypical presentations. While our data show that urinalysis contributed to detecting true infections, we emphasize that urinalysis alone should not dictate antibiotic treatment. Clinical judgment, additional signs or symptoms, and culture results must inform the decision-making process to avoid unnecessary antibiotic use and potential harm. We have clarified this balance in the discussion to stress that while urinalysis can be a valuable component of the diagnostic work-up, it should be interpreted within the broader clinical context rather than used as a standalone test.

We believe these clarifications better reflect the diagnostic challenges and nuances of managing NSC in older adults and strengthen our recommendations. The relevant manuscript sections have been updated to reflect these points (see pages 18-20, lines 329-342 and 370-378): ‘Urinalysis, performed more frequently in the intervention group, detected true infections in about 20% of cases. In nearly 10%, antibiotics were prescribed for what appeared to be asymptomatic bacteriuria. While clinical judgment likely informed these decisions, we did not monitor whether other signs of infection were present. Diagnosing a urinary tract infection (UTI) in older adults is complex, particularly in patients with NSC, as presentations are often atypical, such as confusion, falls, or reduced appetite, rather than classic urinary symptoms (24). Genitourinary infections are among the most frequent diagnoses in this group, which may justify performing urinalysis as part of the diagnostic work-up (3, 25). However, given the high prevalence of asymptomatic bacteriuria and the risk of false positives, urinalysis should always be interpreted in the broader clinical context (26). While our findings support routine urinalysis in this group, treatment decisions should not rely solely on these results. Instead, urinalysis should be one component of the overall diagnostic reasoning, helping to assess the likelihood of a UTI. It is also important to wait for culture results and consider alternative explanations for symptoms to avoid unnecessary antibiotic use. […] Our findings showed that 16% of NSC-patients showed significant abnormalities on ECG. This indicates that the diagnostic yield of ECG in this population is relatively high, with the test revealing a potentially important abnormality in approximately 1 out of every 6 patients. Given the often vague and non-specific nature of NSC presentations and the frequent unreliability of history taking in this population, the ECG appears to be a valuable diagnostic tool that can uncover underlying cardiac conditions that might otherwise remain undetected. These results support the routine use of ECG in the initial assessment of patients with NSC, as it may contribute to earlier recognition of serious pathologies and guide further diagnostic or therapeutic steps.’

Comment 1.2: You may feel that the suggestions for further analysis of additional outcomes made by Reviewer 1 would best be considered in a separate paper.

Reply 1.2: We agree that the suggestions for further analysis of additional outcomes raised by Reviewer 1 are valuable. However, given the exploratory nature of the study and the retrospective data collection in the control group, we chose not to include multivariable or regression analyses, as these could imply a level of causal inference that the study is not designed to support.

Comment 1.3: Can you also check the phrasing on lines 166-169 that I found a bit confusing.

Reply 1.3: Thank you for pointing this out. We have revised the phrasing on lines 166–169 to enhance clarity and readability. See page 13, lines 216-221. ‘Although glucose and alanine aminotransferase (ALAT) already had high completion rates in the control group, they were performed significantly more often in the intervention group (ALAT 98.6% vs. 93.9%, p < 0.001; glucose 99.6% vs 93.9%, p=0.001)’

Comment 1.4: The references also need reviewing to ensure they are complete - I spotted missing page numbers (ref 1), incorrect first author (ref 19) and lack of consistency of citing full or abbreviated journal names (e.g. refs 1 & 2). Has reference 12 now ben published?

Reply 1.4: Thank you for your careful review of the references. We have thoroughly checked and updated all references to ensure completeness and consistency. Page numbers have been added where missing (e.g., ref 1), author details have been corrected (e.g., ref 19), and journal names have been made consistent throughout. Reference 12 has now been published, and the citation has been updated accordingly.

Reviewer #1:

This study addresses an important and timely issue in emergency medicine, optimizing the diagnostic approach for patients presenting with non-specific complaints. Given the growing number of older and medically complex patients in the ED, the evaluation of structured diagnostic pathways is highly relevant. The authors tackle a clinically significant challenge and contribute to an area where evidence is still evolving, making this research both valuable and of broad interest to clinicians and researchers alike.

Dear reviewer,

We thank you for your effort to assess our manuscript. We greatly appreciate the constructive insights that have contributed to its improvement.

In response to your and the reviewers’ comments, we have revised the manuscript accordingly, with all changes highlighted using track changes. We have also provided a detailed, point-by-point reply addressing each of your concerns.

In particular, we have expanded the methods section to enhance this article as a stand-alone piece. We added details about the participating hospitals, emergency departments, and the criteria for nonspecific complaints (NSC) to ensure the study is fully self-contained and its results can be interpreted in light of these characteristics. Additionally, baseline characteristics were included to provide better insight into the study population.

We took note of the insightful comments regarding our recommended diagnostic work-up. Specifically, concerning the role of ECG, we acknowledge that a 16% rate of clinically significant abnormalities is notable and supports the routine use of ECG in this population. Regarding urinalysis, we appreciate the complexity of diagnosing urinary tract infections (UTIs) in older adults, especially those with NSC. While urinalysis contributed to detecting true infections, we emphasize that a positive urinalysis alone should not lead to antibiotic treatment without corroborating clinical signs or symptoms. We have clarified that urinalysis should be integrated with comprehensive clinical assessment and culture results to avoid overtreatment and unnecessary antibiotic use. This nuanced approach better reflects current evidence and clinical practice challenges in this population.

We believe that the revised manuscript is now suitable for publication in European Geriatric Medicine.

On behalf of all authors,

Marleen van der Velde

Comment 2.1: The aim of the study, to assess feasibility, identify unnecessary tests, and evaluate the added value of the NSC care pathway, is clearly stated and relevant. However, the statistical approach, while appropriate for descriptive comparisons (i.e., counts, percentages, chi-square/Fisher’s exact tests), appears limited in its ability to fully address these broader evaluative goals. To strengthen the analysis, the authors might consider including:

- Multivariable analysis to adjust for potential confounders (age, sex, comorbidity, triage level) when comparing testing patterns or outcomes.

- Logistic regression to assess whether specific diagnostic tests were independently associated with relevant clinical outcomes (diagnosis accuracy, admission, return visits).

- If resource utilization is a focus, cost-effectiveness or time-to-diagnosis analyses could also enhance the study’s impact. Did it speed up decisions? Shorten length of stay in the ED? Reduced unnecessary admissions?

Adding these elements could provide a more robust evaluation of the pathway’s effectiveness and clinical relevance beyond adherence and test frequency.

Reply 2.1: We appreciate the thoughtful suggestions to strengthen the analytical approach. However, the primary aim of this study was to explore the feasibility and practical implementation of the NSC care pathway, with a focus on descriptive evaluation of test use, adherence, and basic clinical outcomes. Given the exploratory nature and the study design—particularly the retrospective data collection in the control group—we chose not to include multivariable or regression analyses, as these could imply a level of causal inference that the study is not designed to support. We agree that further research using prospective or interventional designs would be well-suited to address these important questions regarding clinical effectiveness, resource use, and cost-effectiveness.

Comment 2.2: The authors present the background and current knowledge gaps clearly and effectively. The introduction outlines the clinical relevance of the topic and highlights important limitations in existing research, particularly regarding diagnostic challenges in patients with non-specific symptoms. This framing makes the rationale for the study compelling and underscores the potential contribution of the current work to the field of geriatric emergency care.

Reply 2.2: Thank you for your positive feedback. We are pleased to hear that the introduction effectively presented the clinical relevance and research gaps, and that the rationale for our study was clear.

Comment 2.3: The manuscript refers to a previous analysis, which may limit its clarity as a stand-alone article. I recommend rephrasing or expanding this section to ensure the study is fully self-contained and understandable without requiring access to prior publications.

Reply 2.3: Thank you for this suggestion. We have revised the method section to reduce reliance on the previous study and to ensure that the manuscript stands on its own. Additional context and clarifying details have been added, see page 5-6, lines 99-123. ‘The study included patients aged 70 and older who presented to the ED with NSC and met specific inclusion criteria. These criteria included referrals by general practitioners or elderly care physicians during weekday working hours, or triage in the ED with NSCs requiring ED admission. NSC were divided into five referral categories: 1) somatic problems, such as weakness, not feeling well, change in nutritional status or unexplained weight loss; 2) an increased demand of care, such as loss of independency, a necessity for starting home-care or for change in the living situation (such as admission to a care-home), not indicated previously; 3) cognitive problems, such as disorientation, changes in behavior, abrupt cognitive decline; 4) a decline in functional status, such as loss of mobility; and 5) unexplained falls, not related to extrinsic factors.

NSC care-pathway

The detailed trial protocol, including specifics of the care-pathway, has been published previously (7, 12). Before the implementation of the care pathway, patients received standard care. Upon arrival, they underwent triage, followed by history taking and physical examination. Frailty screening was not routinely performed, and there was no standardized approach for screening across the multiple domains of the comprehensive geriatric assessment (CGA); both were left to the discretion of the attending physician. Diagnostic tests were ordered according to the physician’s and nursing staff's judgment.

In contrast, the care pathway offered standardized care for patients with NSC (7). Upon their arrival at the ED, patients were triaged, and assessed for frailty in both cognitive as functional domains using the APOP-screener (13). In addition to solely somatic history taking, physicians conducted screening across various domains in line with the CGA (14). NSC-patients received standard diagnostic measurements, including an electrocardiogram, urinalysis, chest radiography, bladder ultrasound and comprehensive blood tests (table 1). Further diagnostic tests could be performed according to the physician’s choice.’

Comment 2.4: I recommend expanding the "Design and Setting" section to provide additional context about the two hospitals. For example, indicating whether the hospitals are located in urban or rural areas, whether their EDs operate 24/7, and including approximate bed counts or annual ED visit volumes would help readers better understand the study setting and its generalizability.

Reply 2.4: Thank you for this helpful suggestion. We have expanded the "Design and Setting" section to include additional context about both hospitals

---

## [Decision Letter · Decision Letter 1]

6 Aug 2025

Dear Dr. van der Velde,

Thank you for submitting your manuscript to PLOS ONE and for your careful revisions. **You will see below that the reviewer has just spotted one inconsistency.** Therefore, after careful consideration, we feel that your paper has merit but does not yet fully meet PLOS ONE’s publication criteria as it currently stands. Therefore, we invite you to submit a revised version of the manuscript that addresses the point raised during the review process.

We look forward to receiving your revised manuscript.

Kind regards,

Antony Bayer

Academic Editor

PLOS ONE

Journal Requirements:

Reviewers' comments:

Reviewer's Responses to Questions

**Comments to the Author**

Reviewer #2: (No Response)

2. Is the manuscript technically sound, and do the data support the conclusions?

Reviewer #2: Yes

3. Has the statistical analysis been performed appropriately and rigorously?

Reviewer #2: Yes

4. Have the authors made all data underlying the findings in their manuscript fully available?

Reviewer #2: Yes

5. Is the manuscript presented in an intelligible fashion and written in standard English?

Reviewer #2: Yes

Reviewer #2: Many thanks for your updates and revised recommendations, I feel this is a good summary of your outcomes and that the revised document draws sensible conclusions. I would like however if the phrase "true infections" (line 341) could be altered with regards to the positive urine cultures - the 19.9% of cultures with a pathogen identifed represent bacturia only and can only be classed as true infections if they are correlated with symptoms (as described in the revised text). Could this phrase please be amended to reflect this)

**Do you want your identity to be public for this peer review?** For information about this choice, including consent withdrawal, please see our Privacy Policy

Reviewer #2: No

---

## [Author Response · Author response to Decision Letter 2]

7 Aug 2025

Dear Dr. Bayer,

Thank you for your time and effort in reviewing our manuscript. We have carefully addressed the inconsistency highlighted by the reviewer and revised the sentence accordingly. Please see page 18, lines 332–334:

“Urinalysis, performed more frequently in the intervention group, detected bacteriuria in about 20% of cases. In nearly 10%, antibiotics were prescribed for what appeared to be asymptomatic bacteriuria.”

We believe that the updated manuscript now meets the standards for publication in PLOS One.

On behalf of all authors,

Marleen van der Velde

Reviewer #2: Many thanks for your updates and revised recommendations, I feel this is a good summary of your outcomes and that the revised document draws sensible conclusions. I would like however if the phrase "true infections" (line 341) could be altered with regards to the positive urine cultures - the 19.9% of cultures with a pathogen identifed represent bacturia only and can only be classed as true infections if they are correlated with symptoms (as described in the revised text). Could this phrase please be amended to reflect this)

Reply: Thank you for your time and effort in reviewing our manuscript. We have carefully addressed the inconsistency highlighted by you and revised the sentence accordingly. Please see page 18, lines 332–334: “Urinalysis, performed more frequently in the intervention group, detected bacteriuria in about 20% of cases. In nearly 10%, antibiotics were prescribed for what appeared to be asymptomatic bacteriuria.”

We believe that the updated manuscript now meets the standards for publication in PLOS One.

On behalf of all authors,

Marleen van der Velde

---

## [Editor Report · Decision Letter 2]

11 Aug 2025

Evaluation of diagnostic measurements in patients with non-specific complaints: a secondary analysis after implementation of a care-pathway in the Emergency Department

PONE-D-25-21937R2

Dear Dr. van der Velde,

We’re pleased to inform you that your manuscript has been judged scientifically suitable for publication and will be formally accepted for publication once it meets all outstanding technical requirements.

Kind regards,

Antony Bayer

Academic Editor

PLOS ONE
---

## [Editor Report · Acceptance letter]

PONE-D-25-21937R2

PLOS ONE

Dear Dr. van der Velde,

I'm pleased to inform you that your manuscript has been deemed suitable for publication in PLOS ONE. Congratulations! Your manuscript is now being handed over to our production team.

Kind regards,

on behalf of

Professor Antony Bayer

Academic Editor

PLOS ONE